# Automatically Identifying Sickness Behavior in Grazing Lambs with an Acceleration Sensor

**DOI:** 10.3390/ani13132086

**Published:** 2023-06-23

**Authors:** Bowen Fan, Racheal H. Bryant, Andrew W. Greer

**Affiliations:** Department of Agricultural Sciences, Faculty of Agriculture and Life Sciences, Lincoln University, Lincoln 7647, New Zealand; bowen_fan@hotmail.com (B.F.); racheal.bryant@lincoln.ac.nz (R.H.B.)

**Keywords:** lipopolysaccharide, accelerometer, health status, grazing lambs, behavior

## Abstract

**Simple Summary:**

Behavioral patterns of grazing lambs associated with sickness were evaluated using a model of infusion with the endotoxin lipopolysaccharide (LPS), which can lead to subclinical symptoms of disease. Acceleration sensors are validated to have the potential to identify behavioral patterns of farm animals, which can indicate a deterioration in health. However, there is limited research on automatic identification of sickness behavior of grazing lambs. In the present study, the commercial ear-mounted CowManager SensOor (Agis, Harmelen, The Netherlands) was used to detect the changes in behavioral patterns of grazing lambs and showed promising potential for accurately identifying the sickness behavior of grazing lambs.

**Abstract:**

Acute disease of grazing animals can lead to alterations in behavioral patterns. Relatively recent advances in accelerometer technology have resulted in commercial products, which can be used to remotely detect changes in animals’ behavior, the pattern and extent of which may provide an indicator of disease challenge and animal health status. The objective of this study was to determine if changes in behavior during use of a lipopolysaccharide (LPS) challenge model can be detected using ear-mounted accelerometers in grazing lambs. LPS infusion elevated rectal temperatures from 39.31 °C to 39.95 °C, indicating successful establishment of an acute fever response for comparison with groups (*p* < 0.001). For each of the five recorded behaviors, time spent eating, ruminating, not active, active, and highly active, the accelerometers were able to detect an effect of LPS challenge. Compared with the control, there were significant effects of LPS infusion by hour interaction on durations of eating (−6.71 min/h, *p* < 0.001), inactive behavior (+16.00 min/h, *p* < 0.001), active behavior (−8.39 min/h, *p* < 0.001), and highly active behavior (−2.90 min/h, *p* < 0.001) with a trend for rumination time (−1.41 min/h, *p* = 0.075) in lambs after a single LPS infusion. Results suggest that current sensors have the capability to correctly identify behaviors of grazing lambs, raising the possibility of detecting changes in animals’ health status.

## 1. Introduction

Disease state may alter animal behavior. Monitoring changes in behavior associated with disease may provide a tool through which such infections can be rapidly detected and, when combined with automated sensing technologies, may allow for remote and real-time monitoring of the welfare state of animals. Many ruminant diseases, including mastitis and respiratory tract or gastrointestinal infections, are caused by Gram-negative bacterial infection, where the observed symptoms are the consequence of host response to a structural cell wall component of Gram-negative bacteria, lipopolysaccharide (LPS) [1]. Further, LPS may stimulate the acute phase response of infection, resulting in the release of pro-inflammatory cytokines and a reduction in appetite and growth without inducing pathological disease [2], making administration of purified LPS an effective model for bacterial infections and determining associated changes in host behavior [1,3,4,5,6,7]. Feed intake of lambs and cows is frequently reduced by LPS infusion. Under induced LPS challenge, lambs displayed symptoms of anorexia for up to 48 h with a 50% reduction in feed intake [8], while feed intake of lambs decreased for up to 48 h after intravenous injection of LPS [9]. Feed intake of heifers was suppressed at 4 h after LPS infusion with cumulative feed intake reduced by 33% at 24 h [1]. Similarly, a single dose of LPS with intra-mammary infusion for 12 h reduced dry matter intake (DMI) of dairy cows by 19% [10]. Further, LPS infusion decreased eating duration and ruminating duration of dairy cows with both chewing time and number of chews reduced based on the accelerometer data from RumiWatch sensors [10]. DMI of lactating cows showed a linear decrease with increasing administration of 0.0, 0.5, 1.0, and 1.5 μg LPS/kg BW for 8 h with an associated decrease in milk production for up to two days [6], while cows with an acute mastitis model induced by infusion with 1 μg of LPS had greater stepping frequency, longer standing time and bouts, less lying time, and a longer time eating silage [11].

Changes in behavioral activities in farm animals can be used as an indicator of welfare [12]. Implementation of behavioral monitoring programs with early detection systems against animal disease would enable the adoption of effective control measures and improve animal health and well-being outcomes. However, manual monitoring of grazing animals with direct observation is time-consuming, labor intensive, and impractical. Automatic sensor technologies can be implemented as a valuable management system for continuously monitoring behavioral activities of animals at pasture. In a recent review of accelerometers for detection of animal health changes, there were a number of sensors which were sensitive to changes in behavior and which have been validated to detect the behavioral patterns of animals exposed to sickness [13]. Moreover, various sensors such as GPS, heart rate monitors and echocardiograms, motion sensors, jaw and bite sensors, and contact loggers, have been validated and used in lambs [14]. Accelerometers mounted on and around the head were effective for identifying a range of behaviors. Ear-attached or collar-mounted accelerometers have high accuracy of identifying eating, ruminating, and other behaviors of cattle and lambs, while leg-mounted acceleration sensors can differentiate their motion patterns such as standing, lying, and walking. Among the available accelerometer sensors, the CowManager SensOor system has been widely used to classify behaviors of dairy cattle and monitor their health status. For instance, CowManager SensOor has been tested with good validation performance for eating, ruminating, inactive, active, and highly active behaviors [15,16,17,18] and to measure behavioral patterns for predicting estrus [19] and calving onset [20]. Furthermore, CowManager SensOor was used to evaluate the impact of LPS injection on behaviors of beef steers with the results indicating LPS infusion reduced the duration of eating, ruminating, and both active and highly active behaviors, and increased the time of inactive behaviors [21]. However, there is limited research on the detection of behavioral changes in lambs exposed to sickness challenges via the commercial CowManager SensOor system.

The purposes of the present study were to (1) validate the data of behavioral durations from the acceleration sensor CowManager SensOor in comparison with visual observation, (2) and measure behavioral activity levels in grazing lambs on pasture through CowManager SensOor in response to experimental exposure to a single dose of LPS as an infection model.

## 2. Materials and Methods

### 2.1. Experimental Site and Design

Animal ethical approval for the procedure in this study was provided by the Lincoln University Animal Ethics Committee #2021-11. The experiment was carried out between 12 July 2021 and 4 August 2021 at the Lincoln University Research Farm (Latitude 43°38′35″ S, Longitude 172°27′16″ E). The experimental design was a 2 × 2 Latin square with two intravenous injections of 0 or 0.5 μg/kg BW of endotoxin (*Escherichia coli* Serptype 055: B5 lipopolysaccharide, LPS, Sigma, St. Louis, MO, USA) within two periods.

### 2.2. Animals and Management

Animals used in the study were twenty female Coopworth lambs of 8–10 months old (mean live weight = 38.63 ± 2.04 kg) grazed on a permanent perennial ryegrass and white clover pasture. At recruitment on 12 July, all lambs were weighed and fitted with CowManager SensOor ear tags for a week before the experiment. All acceleration sensors were attached to the left ear between half and one third from the head of each lamb. All lambs were grazed together ad libitum as one mob on pastures with unrestricted access to water. After weighing, the lambs were allocated within live weight strata to one of two treatment groups, which respectively received 0 or 0.5 μg/kg.BW of endotoxin (*Escherichia coli* Serptype 055: B5 lipopolysaccharide, LPS, Sigma, St. Louis, MO, USA). A concentration of 5 μg LPS/mL sterile 0.9% saline solution was freshly prepared on the day of challenge. On 19 July, the lambs in LPS group were intravenously injected in a single bolus for while the control ones were intravenously injected with the sterile 0.9% saline solution. A week after the first intravenous infusion of LPS on 26 July, the two groups (*n* = 10/group) were swapped and intravenously administered with 0.5 or 0 μg/kg.BW of lipopolysaccharide, and the experimental procedure was repeated for the second period.

### 2.3. Measurements

#### 2.3.1. Infection Status

Animals were briefly yarded when they received LPS solution before returning to their paddock. Approximately 4–5 h after the intravenous injection of LPS solution when a fever was expected to peak, all lambs were briefly yarded again to measure rectal temperature using a clinical digital thermometer (OMRON, MC-341, OMRON HEALTHCARE, Kyoto, Japan) and then returned to the paddock of about 0.75 ha of permanent semi-natural pastures.

#### 2.3.2. Herbage Mass

The herbage mass was measured prior to the commencement of the trial by cutting 4 quadrats, each 0.2 m^2^ to the ground level in the sward. Herbage samples were oven-dried at 65 °C for 48 h and weighed. The samples were analyzed for concentrations of crude protein (CP), neutral detergent fiber (NDF), and in vitro organic matter digestibility (OMD) for calculating metabolizable energy (ME) concentration.

#### 2.3.3. Behavior

The behaviors of all lambs were simultaneously monitored through visual observation and via the ear-attached sensors over a three-day period, prior to and post-intravenous infusion of LPS for validation. For visual observation, four lambs and three lambs, respectively, were randomly selected and individually marked for visual identification. Direct visual observations were recorded by three trained observers (1, 2, and 3) who were randomly assigned lambs to monitor. Before the trial, all observers discussed the definition of different behaviors until an agreement was reached. Each animal’s behaviors were categorized by sensors which were previously calibrated for cattle. These behaviors from sensor recordings are condensed into five categories: “ruminating”, “eating”, “not active”, “active”, and “highly active”, which are defined as “REGURGITATING a bolus and chewing the cud while lying, standing, walking or moving her head and jaw in a circular motion and then swallowing the masticated cud”, “The muzzle was close to or near the ground and ripped the forage and chewed it with eating jaw movements”, ”Standing at resting or lying without jaw movement or further activity”, “Standing or lying with minor body and/or head movement”, and “Walking and/or moving head and body clearly”, respectively. Highly active behavior was classified into the category of “active” during validation performance. During visual observation period, each minute was classified as only one of four behaviors (ruminating, eating, inactive, or active), which were mutually exclusive. The observers were positioned outside the fence of the pasture enclosure where the lambs grazed in order to not disturb the animals despite them being habituated to human presence, without an unobstructed view on the individual lamb. During the two periods, observer 1 recorded the behaviors of four lambs, while observer 2 and observer 3 recorded the behaviors of three lambs during the first period and the second period, respectively. The activity of each lamb was recorded every minute for 60 consecutive minutes between 09:00 am and 16:00 pm for six days during the whole trial (20, 21, 23, 28, 29, and 30 July).

The data from one lamb were removed from the analysis due to its death on day one of experiment. The datasets were statistically analyzed via Genstat software (VSN International, 19th Edition), using mixed models with repeated measures. A paired sample *t*-test was conducted to determine the difference in behavioral patterns of sensor recordings and visual observation for validation. The duration of each behavior category in lambs recorded via CowManager SensOor after LPS challenge was compared as dependent factors in a repeated measure mixed model with treatment and hour as fixed effects and individual animal as a random effect. The diurnal behavioral durations were compared using the REML with the date and animal within the hour and treatment as the fixed model, and with the date and hour within animal as the random model for investigation. The autoregressive model for a continuous time covariate was fitted to account for the time autocorrelation. Statistical significance of the effects for all the analyses was considered at *p* < 0.05.

## 3. Results

### 3.1. Rectal Temperature

The mean rectal temperature (mean ± SE) recorded in LPS-treated animals was 39.95 ± 0.11 °C, which was higher compared with 39.31 ± 0.08 °C in the control animals (*p* < 0.001).

### 3.2. Herbage Mass

All lambs had similar nutritional conditions with sufficient pasture herbage of high nutritional quality. The concentration per kg of dry matter herbage was 154.43 ± 2.09 g CP, 501.98 ± 12.61 g NDF, and 11.30 ± 0.13 MJ ME.

### 3.3. Validation for the Data between Visual Observation and Sensor Detection

Table 1 presents average time spent performing each behavior, the analysis of sensor recordings, and visual observation. There was good agreement between visual observations and sensors for active and not active behavior, but poor agreement with eating and ruminating time (Figure 1). Compared with visual observation results, sensor recordings showed lower eating time (*p* = 0.002) and ruminating time (*p* = 0.043), higher active time (*p* < 0.001) and no difference in time spent not active (*p* = 0.23). Regression analysis (Table 1) indicated high linear relationships between visual observations and sensor recordings for time spent not active (R^2^ = 0.75, *p* < 0.001) and active (R^2^ = 0.75, *p* < 0.001) with a poor association for ruminating time (r = 0.08, *p* = 0.48). The intercept for time spent eating, ruminating, and active were all significantly different from zero (*p* < 0.001), while the slope for time spent eating, not active, and active were all significantly different from zero (*p* < 0.001).

### 3.4. Effects of LPS Challenge on Activity Patterns

#### 3.4.1. Eating Duration

Compared with the control group, there was a significant effect of hour (*p* < 0.001) and LPS infusion by hour interaction (*p* < 0.001) on mean eating duration of lambs. Over 3 h after LPS challenge, the eating duration of affected lambs was reduced by 6.71 min/h compared with the control animals (Figure 2a).

#### 3.4.2. Ruminating Duration

Significant effects of hour (*p* < 0.001) and LPS infusion by hour interaction (*p* < 0.001) on ruminating duration of lambs were found in comparison with the control group (Figure 2b). Compared with the control, there was a trend for decreased rumination time within 4 h after LPS infusion (−1.41 min/h) (*p* = 0.075).

#### 3.4.3. Inactive Duration

There were significant effects of hour (*p* < 0.001) and LPS infusion by hour interaction (*p* < 0.001) on inactive behavior duration compared with the control group (Figure 2c). Compared with the ones without LPS infusion, LPS infusion within 4 h increased inactive duration by 16.00 min/h.

#### 3.4.4. Active Duration

There were effects of hour (*p* < 0.001) and LPS infusion by hour interaction (*p* < 0.001) on active behavior duration (Figure 2d). Compared with the control, the active duration was decreased by 8.39 min/h due to LPS infusion within 4 h.

#### 3.4.5. Highly Active Duration

There were significant effects of hour (*p* < 0.001) and LPS infusion by hour interaction (*p* < 0.001) on highly active duration in comparison with the control group (Figure 2e). LPS infusion increased the highly active duration of affected within 3 h by 2.90 min/h.

## 4. Discussion

The aims of this study were to identify the behavioral patterns of grazing lambs through visual observation and sensor-derived recordings for validation and to quantify and contrast the behavioral changes in grazing lambs after LPS infusion. In this study, approximately four hours after a single LPS infusion, rectal temperature was significantly increased in lambs, indicating a fever response. Previous studies have also reported increased rectal temperature between 2 and 8 h after intravenous injection of LPS [22]. Rectal temperature of beef steers was increased 6 h post-injection of LPS [21], and rectal temperature of dairy cows were at peak level 6 h after LPS infusion [10]. There was also a mild increase in rectal temperature of Holstein × Jersey heifers 100 min after LPS infusion with a dose of 2 μg/kg [1].

Overall, there was poor correlation between visual and sensor data for eating, ruminating, and inactivity. There are several reasons for the discrepancies, not least being that despite them being a flock from a research farm and thus acclimated to human presence, the natural flighty nature of lambs makes visual recording in extensive situations more difficult, particularly the detection of jaw movement from a distance, and many inactive and rumination activities. With respect to eating behavior, the poor correlation between visual and sensor was unexpected, but may reflect the fact the CowManager SensOor algorithms were initially validated in cattle which have a considerably different grazing motion than sheep. By contrast, ruminating time and eating time were similar for visual observation compared with the sensor data, while the time being inactive was greater and active behavior had a trend to be lower for visual observation compared with the sensor recording [23]. However, the ruminating time was also shown not to be different between sensor and visual observation; furthermore, and eating time while resting recorded by the sensor was higher, and the time being active recorded by the sensor was lower compared with visual observations [16]. Further, a negligible correlation for eating time (*r* = 0.27) between visual observation and the sensor recordings was reported [24]. It is possible that the lambs were displaying other sorts of activities (such as exploring, rising, lying down, head movement, and social interaction) that were not categorized within the current study or sensor recordings based on the algorithms used [23]. Further, the changes in active behavior and high activities may be difficult to be detected by the CowManager SensOors, which only record complex patterns of ear movement through an accelerometer for classifications of detailed activities [16,23]. It is difficult to correctly classify complex ear movement patterns associated with eating and active behavior [16], and when cows graze with a posture of standing or walking, the grazing behavior may be regarded as both active and eating [25]. Moreover, this highlights the challenge of utilizing commercially available technologies, especially where the algorithms are calibrated in other species. With that in mind, there are limited options for a commercial operator to obtain sheep-specific devices. Even if they existed, there is a balance between the likelihood of greater farmer uptake and the impact of studies compared with the limited control over setting appropriate thresholds for the recording of different levels of activity that may be required for scientific purposes.

Ear tag sensor data showed LPS injections had an immediate effect (within 4 h) of reducing the activity of the animals and activity categorized as eating was also reduced. These results are in agreement with previous studies, whereby LPS infusion has been shown to cause lethargy and inappetence. Steers receiving intravenous injection of LPS spent less time eating than the control based on the data collected by the ear tag-based CowManager SensOor [21]. In the current study, the reduced eating duration of lambs exposed to LPS infusion was temporal, peaking 1 h after treatment before recovering at 4 h post-treatment. This effect was shorter in duration than expected, as previous studies have shown LPS to reduce intake for longer periods in both sheep [8] and cattle [26]. However, grazing time is not always a good indicator of intake [27] as animals may use compensatory behaviors to maintain intake [28]. Bulls exposed to LPS challenge consumed less feed by a slower rate of eating, instead of reducing feeding time [29]. It was, however, expected that eating time would be reduced following LPS challenge. For instance, there was a significant reduction in feed intake of Holstein × Jersey heifers 4 h after LPS infusion [1]. A decreased duration of hay eating (LPS vs. control 23.11 ± 6.93 min vs. 31.52 ± 7.54 min) of dairy calves was found [22]. Unfortunately, in the current study the correlation between sensor and visual data for intake was low, so limited conclusions can be drawn with regards to the ability of the sensors to detect the impact of LPS on eating the behavior of lambs.

Reduction in ruminating behavior following LPS challenge was expected, but was not consistently shown in the current study. In previous research using a 4 h video recording, ruminating duration of dairy calves was reduced by LPS infusion compared with the control (6.42 ± 3.69 min vs. 24.57 ± 6.64 min, respectively) [22]. Intravenous injection of LPS induced less ruminating time of steers than the control based on the sensor data [21]. In another study with goats, which used 11 h video recording from 9 AM to 8 PM, observers found a significant decrease in cumulative ruminating duration when exposed to LPS infusion (133.5 ± 9.4 min) compared with the control (195.0 ± 11.8 min) [30]. Similarly, in cows receiving the LPS infusion, ruminating duration over 8 h was reduced by 35% [10], which was also associated with reduced total chewing time as well as number of boli, chews per bolus, and chews per minute. Although unexpected, presumably the lack of effect of LPS detected on rumination in the current study may simply reflect the poor accuracy of the sensor to detect rumination in lambs as indicated in the comparison with visual observations (Figure 1), once again limiting conclusions that can be drawn.

Pronounced differences in activity were evident from the sensor recordings. These observations were similar to those of previous studies that reported increased time spent being inactive and the decreased time spent being active under the LPS challenge. For example, the durations of time spent lying and standing inactive in dairy calves were increased and the frequency of self-grooming was decreased due to LPS injection [22]. LPS infusion significantly reduced the cumulative number of goats’ self-grooming [30]. Steers receiving intravenous LPS infusion spent less time being highly active, and more time being inactive than the control, based on the sensor data [21]. Reduction in activity was presented in beef cattle approximately 10 h after the first LPS challenge [31]. Furthermore, LPS infusion decreased activity of pigs during the first hours after injection [32]. A decrease in activity has also been recorded in sparrows [33] and mice [34,35,36,37] following an LPS challenge. Reduced activity related to fever, usually as a mechanism of energy conservation, may also be associated between the LPS-induced depression and the activation of pro-inflammatory cytokines as part of the immune reaction [38,39]. As such, active and inactive both appear to be sensitive measures of animal sickness behavior, which can be assessed with the use of the current sensors.

The CowManager SensOors provide a feasible approach to help improve early detection of animal diseases for monitoring behavioral patterns of lambs at pasture without human observations. Although in a practical context the detection of disease may be limited to times of the day when animals may be expected to exhibit the indicator behavior. With this in mind, the timing for the administration of LPS was deliberately chosen to fall in the midst of their morning grazing to provide the maximum chances of seeing an effect. Given that LPS is a short-term insult to the animal’s behavior patterns, it remains to be investigated if similar effects can be detected at other times of the day when animals may not normally be exhibiting the behaviors measured here. An example of this is during studies regarding chronic challenges of alkaloids from ryegrass endophytes, researchers have observed changes in behavior during periods of rest in the evening but not during the day [40]. Nevertheless, the current study demonstrates the analysis of the collected datasets from the accelerometer sensor and provides a practical detection criterion for identifying the behavioral changes induced by subclinical infections, which allow for more effective treatments against illness.

## 5. Conclusions

The CowManager SensOor has the capability of capturing data corresponding to the activity levels of grazing lambs infected with LPS at pasture, and can be used to identify the eating, ruminating, inactive behaviors, and active behaviors. Moreover, when exposed to an immunological insult through the use of LPS as a challenge model, changes in the activity levels of animals, most notably those classified as inactive and active, provide sensitive measures of animal well-being.

## Figures and Tables

**Figure 1 animals-13-02086-f001:**
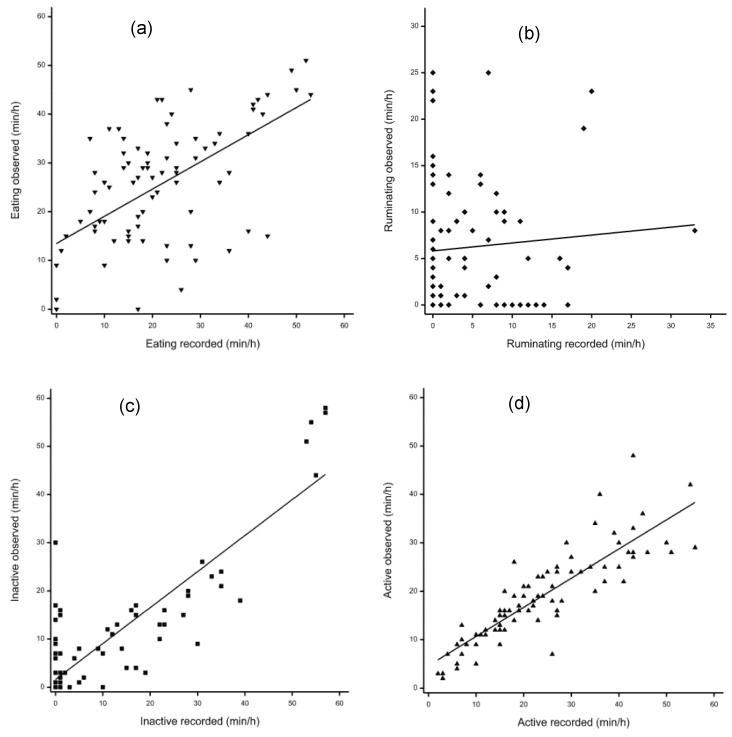
Relationship between CowManager SensOor recordings and visual observation. (**a**) Eating duration (⯆); (**b**) ruminating duration (⯁); (**c**) inactive duration (⯀); (**d**) active duration (⯅).

**Figure 2 animals-13-02086-f002:**
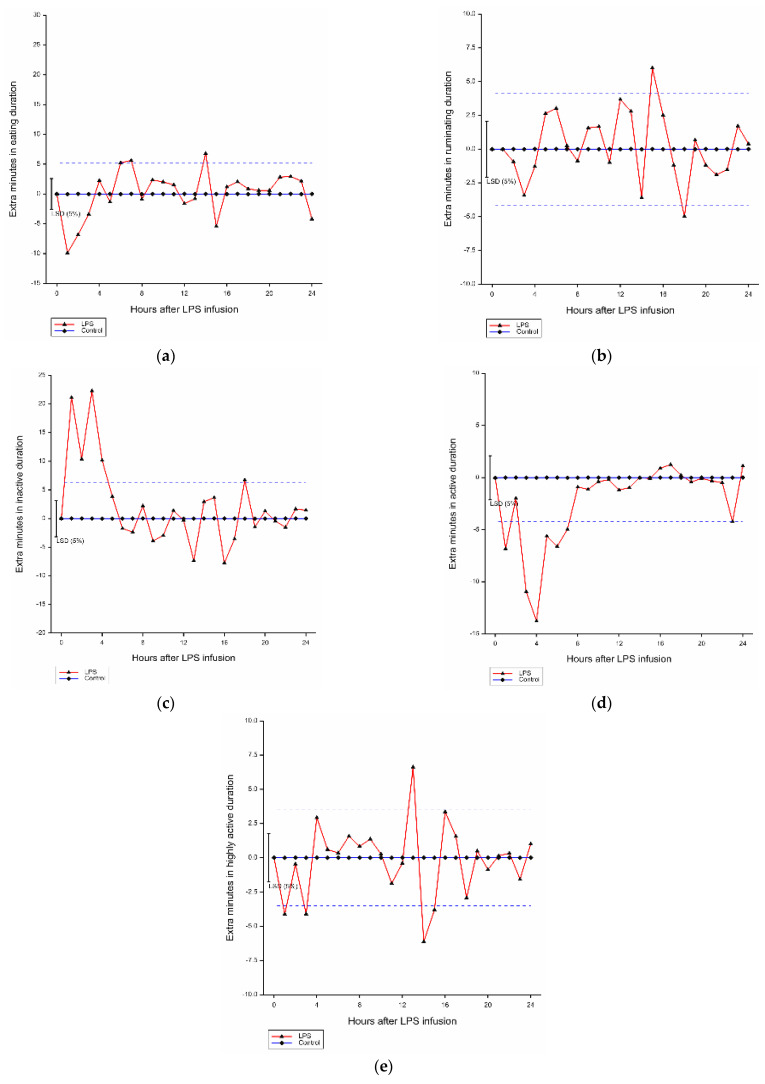
Mean hourly duration of each behavior in grazing lambs after LPS infusion within 24 h (difference from control). (**a**) Average hourly eating duration; (**b**) average hourly ruminating duration; (**c**) average hourly inactive duration; (**d**) average hourly active duration; (**e**) average hourly highly active duration. LSD: least significant difference, blue dashed line: LSD from the control.

**Table 1 animals-13-02086-t001:** Average hourly duration (mean ± SEM, min/h) of each behavior in lambs for sensor recordings versus visual observation with *p*-values for the 2-sided paired *t*-test and Pearson correlation co-efficient (r) with *p*-values and regression analysis results between sensor recordings and visual observation for time spent for each behavior category.

	Eating	Ruminating	Inactive	Active
Visual observation	25.39 ± 1.35	6.18 ± 0.73	9.33 ± 1.47	19.08 ± 1.03
Sensor recordings	21.38 ± 1.43	4.23 ± 0.66	10.36 ± 1.70	23.99 ± 1.48
*p*-value for *t*-test	0.002	0.043	0.23	<0.001
r	0.59	0.08	0.87	0.86
*p*-value for r	<0.001	0.48	<0.001	<0.001
R^2^	0.35 (*p* < 0.001)	0.006 (*p* = 0.48)	0.75 (*p* < 0.001)	0.75 (*p* < 0.001)
Slope (SEM, *p*)	0.56 (0.08, *p* < 0.001)	0.085 (0.12, *p* = 0.48)	0.75 (0.05, *p* < 0.001)	0.60 (0.04, *p* < 0.001)
Intercept (SEM, *p*)	13.47 (2.12, *p* < 0.001)	5.82 (0.90, *p* < 0.001)	1.60 (0.88, *p* = 0.07)	4.65 (1.06, *p* < 0.001)

Visual observations and CowManager SensOor recordings were compared on an hourly basis with results conducted using a 2-sided paired *t*-test. The regression model between visual observation (Y-axis) and CowManager SensOor (X-axis) measurements (min/observation) are presented with the coefficients of determination (R^2^), the slopes, and the intercepts with standard errors of the mean (SEM) and *p*-value. The significance level for *p*-value was set at 0.05.

## Data Availability

Data are included within the article.

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
