# Peer review of "Automatically Identifying Sickness Behavior in Grazing Lambs with an Acceleration Sensor"

_animals, 2023, doi:10.3390/ani13132086_

Round 1

Reviewer 1 Report

This is a good paper based on a sound study.  It addresses an important topic.  The paper is generally well written.  I have a few suggestions below that should improve the paper.

Lines 275-278.  This point need more discussion.  The previous part of the paragraph could be summarized.  Remind the reader that the algorithm used for the Cowmanager SensOor was developed for cows not lambs. Using accelerometers and machine learning to predict behavior from observations is often very accurate for sheep. For example, Fogarty was able to predict activity with an accuracy over 90%.  Commercial products that rely on one commercially development algorithm to predict behavior is not surprisingly less accurate than using accelerometers and using machine learning to predict behavior.

Lines 307-308.  Clearly state that the inability of the Cowmanager SensOor to accurately predict rumination is likely the reason that there was no detectable decline in rumination when challenged with LPS injection.   The current sentence is not clear on this point.

The LPS injection clearly decreased activity and correspondingly increased inactivity.  However, this is only apparent when lambs would normally be active (such as grazing). Livestock behavior including lambs has a diurnal pattern.  This is verified in this study because activity and inactivity varied by hour (hour was significant). The discussion should mention that detection of disease should be based on a reduction in activity (and increase in inactivity) during times of the day that lambs are normally active such as during the normal morning and evening grazing bouts.  Accelerometers will not be able to detect a decrease in activity and increase in inactivity during periods when the lambs are normally not active, such as night. For example, Trieu et al. (2022) detected a change in activity of lambs with ryegrass staggers during the daylight but not at night.  I think it is important to note that a decrease in activity from disease (or LPS injection) is most notable when the animal is normally active such as at feeding or grazing periods.

To further illustrate the above point, I suggest changing Figure 2 so that you show a full 24-hour day and the corresponding normal diurnal activity pattern of the control sheep and the change in the diurnal pattern of the LPS injected lambs. I think graphing the activity pattern of both control and LPS injected lambs on the same figure would be more informative than showing the deviation of LPS lambs from the controls.

Fogarty, ES, Swain, DL, Cronin, GM, Moraes, LE, Trotter, M (2020) Behaviour classification of extensively grazed sheep using machine learning. Computers and Electronics in Agriculture 169, 105175.

Trieu, LL, Bailey, DW, Cao, H, Son, TC, Scobie, DR, Trotter, MG, Hume, DE, Sutherland, BL, Tobin, CT (2022) Potential of Accelerometers and GPS Tracking to Remotely Detect Perennial Ryegrass Staggers in Sheep. Smart Agricultural Technology 100040.

Author Response

Reviewer 1

This is a good paper based on a sound study.  It addresses an important topic.  The paper is generally well written.  I have a few suggestions below that should improve the paper.

We thank the reviewer for the positive words and thank them for the suggestions for improvement.

Lines 275-278.  This point need more discussion.  The previous part of the paragraph could be summarized.  Remind the reader that the algorithm used for the Cowmanager SensOor was developed for cows not lambs. Using accelerometers and machine learning to predict behavior from observations is often very accurate for sheep. For example, Fogarty was able to predict activity with an accuracy over 90%.  Commercial products that rely on one commercially development algorithm to predict behavior is not surprisingly less accurate than using accelerometers and using machine learning to predict behavior.

Response:  We thank the review for this suggestion and agree.  We have revisited this paragraph and attempted to make it more pointed and have included more of a focus  on the fact that these algorithms are not designed for sheep, but due to the lack of commercially available products for sheep then alternative compromises are needed to be made.

 Lines 307-308.  Clearly state that the inability of the Cowmanager SensOor to accurately predict rumination is likely the reason that there was no detectable decline in rumination when challenged with LPS injection.   The current sentence is not clear on this point.

Response:  Complied: we have adjusted the sentence to now read ‘Although unexpected, presumably the lack of effect of LPS detected on rumination  in the current study may  simply reflect the poor accuracy of the sensor to detect rumination in lambs as indicated in the comparison with visual observations (Fig 1), once again limiting conclusions that can be drawn.’

 The LPS injection clearly decreased activity and correspondingly increased inactivity.  However, this is only apparent when lambs would normally be active (such as grazing). Livestock behavior including lambs has a diurnal pattern.  This is verified in this study because activity and inactivity varied by hour (hour was significant). The discussion should mention that detection of disease should be based on a reduction in activity (and increase in inactivity) during times of the day that lambs are normally active such as during the normal morning and evening grazing bouts.  Accelerometers will not be able to detect a decrease in activity and increase in inactivity during periods when the lambs are normally not active, such as night. For example, Trieu et al. (2022) detected a change in activity of lambs with ryegrass staggers during the daylight but not at night.  I think it is important to note that a decrease in activity from disease (or LPS injection) is most notable when the animal is normally active such as at feeding or grazing periods.

Complied, we have included a mention of the fact that behaviours would normally need to be exhibited for there to be a difference and we have included the reference of Trieu et al 2002.  The final paragraph now reads:

The CowManager SensOors provides a feasible approach to help improve early detection of animal diseases for monitoring behavioral patterns of lambs at pasture without human observations. Although in a practical context the detection of disease may be limited to times of the day when animals may be expected to normally exhibiting the indicator behavior.  With this in mind, the timing for the administration of LPS was deliberately chosen to fall in the midst of their morning grazing bout to give the maximum chances of seeing an effect. Given that LPS is a short-term insult to the animal’s behavior patterns, it remains to be investigate if similar effects are able to be detected at other times of the day when animals may not normally be exhibiting the behaviors measured here. An example of this is, studies where chronic challenges of alkaloids from ryegrass endophytes have observed changes in behavior during the day but during periods of rest in the evening [40]. Nevertheless, the current study demonstrates the analysis of the collected datasets from the accelerometer sensor provides a practical detection criterion for identifying the behavioral changes induced by sub-clinical infections, which allow for more effective treatments against illness.

To further illustrate the above point, I suggest changing Figure 2 so that you show a full 24-hour day and the corresponding normal diurnal activity pattern of the control sheep and the change in the diurnal pattern of the LPS injected lambs. I think graphing the activity pattern of both control and LPS injected lambs on the same figure would be more informative than showing the deviation of LPS lambs from the controls.

We thank the reviewer for this suggestion and can see their point with regard to the demonstration of the total activity time across the day.  We are happy to change our view on this if the editor insists but for the moment we prefer to leave the display of the graphs in relation to the controls.  The reason for this being that the LPS is a short-term insult to the animals, and the objective of this study was to determine if a change in activity could be detected in sick animals.  As such, a direct comparison of the behaviour relative to a control animal seems to be the most appropriate.

 Fogarty, ES, Swain, DL, Cronin, GM, Moraes, LE, Trotter, M (2020) Behaviour classification of extensively grazed sheep using machine learning. Computers and Electronics in Agriculture 169, 105175.

 Trieu, LL, Bailey, DW, Cao, H, Son, TC, Scobie, DR, Trotter, MG, Hume, DE, Sutherland, BL, Tobin, CT (2022) Potential of Accelerometers and GPS Tracking to Remotely Detect Perennial Ryegrass Staggers in Sheep. Smart Agricultural Technology 100040.

Reference included

Reviewer 2 Report

In general, the authors do a good job of presenting their information in relation to existing literature. Major comments:

Section 2.2.3: Why did the authors choose only 3 or 4 lambs per observer and why did 2 observers monitor blue while only 1 observer monitor green? Were different observers used for different periods  in the crossover design?

Additionally, were the lambs control, lps, or other? If from various treatments, were the treatments balanced among observers?

Were lambs trained or conditioned to the presence of people prior to the experiment?

The rationale regarding discrepancies between visual observation and accelerometers is confusing. Theoretically, in a crossover design, there should be no period effect or the period effect should be tested. The authors speculate in lines 262-264 that weather could have impacted visual vs accelerometer data, but couldn’t weather also have impacted the other variables measured?

Other minor comment:

After a brief search, it appears that there are other articles related to sheep health/production and accelerometers. The authors should consider updating any references/discussion.

The article titled “Automatically identifying sickness behavior in grazing lambs with an acceleration sensor” evaluates various behavior characteristics detected by accelerometer in response to LPS challenge. The article would benefit from an additional review for grammatical errors. There are several errors throughout the article and is beyond the scope of peer reviewers. Third party editing services are available. An example of grammatical errors: consistent use of hours vs. h, end of line 64 has both period and comma, line 81 …the duration s of eating,…, and numerous sentences that can be reworded for clarity.

Author Response

In general, the authors do a good job of presenting their information in relation to existing literature.

Thank you

Major comments:

Section 2.2.3: Why did the authors choose only 3 or 4 lambs per observer and why did 2 observers monitor blue while only 1 observer monitor green? Were different observers used for different periods  in the crossover design?

Thank you for pointing out the need for clarification.  In the revised submission we have removed reference to the blue and green as these potentially unnecessarily add to confusion.  We have also made amendments this section which we hope address the reviewers concerns, but in short yes, there were three observers used throughout the study, one was used every time and the other two interchanged

Additionally, were the lambs control, lps, or other? If from various treatments, were the treatments balanced among observers?

The lambs were randomly allocated across both treatments and observers.  As above, we have modified this section and trust that it is clearer to the reader now.  With this in mind the objective of the visual observation was to validate the ability of the sensors to determine what activity the lamb was undertaking, from this perspective the treatment is not relevant as an LPS treated lamb resting would be expected to have the same behaviour as a control lamb resting.

Were lambs trained or conditioned to the presence of people prior to the experiment?

We have included specific mention of this in the discussion and materials and methods.  The lambs were from a research farm so were used to having humans around – but whether they were completely habituated is unlikely. Care was taken to be as discreet as possible and not interrupt their behaviour during the observation periods, but one can never be certain that their behaviour was not modified in some way.

The rationale regarding discrepancies between visual observation and accelerometers is confusing. Theoretically, in a crossover design, there should be no period effect or the period effect should be tested. The authors speculate in lines 262-264 that weather could have impacted visual vs accelerometer data, but couldn’t weather also have impacted the other variables measured?

We agree, and have removed this statement from the manuscript

Other minor comment:

After a brief search, it appears that there are other articles related to sheep health/production and accelerometers. The authors should consider updating any references/discussion.

Thank you, as above, for reviewer 1, we have included the reference of Trieu et al.  This is a field that is rapidly evolving, although a majority of the more recent manuscripts have a focus on the most appropriate methods of data analyses, which, although very useful in themselves, in the context of the objectives of this study they would detract from the key message.